# Improving community health worker treatment for malaria, diarrhoea, and pneumonia in Uganda through inSCALE community and mHealth innovations: A cluster randomised controlled trial

**Karin Källander** [1,2,3‡]*, **Seyi Soremekun**[4‡], **Daniel LI Strachan**[5,6], **Zelee Hill**[5], **Frida Kasteng**[2,7], **Edmound Kertho**[8], **Agnes Nanyonjo**[2,8], **Guus Ten Asbroek**[9], **Maureen Nakirunda**[8], **Patrick Lumumba**[8], **Godfrey Ayebale**[8], **Benson Bagorogoza**[8], **Anna Vassall**[7], **Sylvia Meek**[1†], **James Tibenderana**[1‡], **Raghu Lingam**[10,11‡], **Betty Kirkwood**[10‡]

1 Malaria Consortium, London, United Kingdom, 2 Department of Global Public Health, Karolinska Institutet, Stockholm, Sweden, 3 Health Programme Group, UNICEF, New York, New York, United States of America, 4 Department of Infection Biology, London School of Hygiene & Tropical Medicine, London, United Kingdom, 5 Institute for Global Health, University College London, London, United Kingdom, 6 The Nossal Institute for Global Health, Melbourne School of Population and Global Health, The University of Melbourne, Victoria, Australia, 7 Department of Global Health and Development, London School of Hygiene & Tropical Medicine, London, United Kingdom, 8 Malaria Consortium, Kampala, Uganda, 9 Amsterdam UMC, location University of Amsterdam, Department of Global Health, Amsterdam Institute for Global Health and Development, Amsterdam, the Netherlands, 10 Department of Population Health, London School of Hygiene & Tropical Medicine, London, United Kingdom, 11 Population Child Health Research Group, School of Women's and Children's Health, University of New South Wales, Australia

‡ KK and SS share first authorship on this work. JT and RL and BK are joint senior authors on this work.
† Deceased.
* Karin.kallander@ki.se

**Data Availability Statement:** Data is available at: Soremekun, S (2022). inSCALE Uganda Impact

## Abstract

The inSCALE cluster randomised controlled trial in Uganda evaluated two interventions, mHealth and Village Health Clubs (VHCs) which aimed to improve Community Health Worker (CHW) treatment for malaria, diarrhoea, and pneumonia within the national Integrated Community Case Management (iCCM) programme. The interventions were compared with standard care in a control arm. In a cluster randomised trial, 39 sub-counties in Midwest Uganda, covering 3167 CHWs, were randomly allocated to mHealth; VHC or usual care (control) arms. Household surveys captured parent-reported child illness, care seeking and treatment practices. Intention-to-treat analysis estimated the proportion of appropriately treated children with malaria, diarrhoea, and pneumonia according to WHO informed national guidelines. The trial was registered at ClinicalTrials.gov (NCT01972321). Between April-June 2014, 7679 households were surveyed; 2806 children were found with malaria, diarrhoea, or pneumonia symptoms in the last one month. Appropriate treatment was 11% higher in the mHealth compared to the control arm (risk ratio [RR] 1.11, 95% CI 1.02, 1.21; p = 0.018). The largest effect was on appropriate treatment for diarrhoea (RR 1.39; 95% CI 0.90, 2.15; p = 0.134). The VHC intervention increased appropriate treatment by 9% (RR

Evaluation Dataset. [Data Collection]. London School of Hygiene & Tropical Medicine, London, United Kingdom. https://doi.org/10.17037/DATA.00002559.

**Funding:** The study was funded by Bill & Melinda Gates Foundation (https://www.gatesfoundation.org/) and awarded to co-author SM (Grant number OPP1002407). The funders had no role in study design, data collection and analysis, decision to publish, or preparation of the manuscript.

**Competing interests:** The authors have declared that no competing interests exist.

1.09; 95% CI 1.01, 1.18; p = 0.059), again with largest effect on treatment of diarrhoea (RR 1.56, 95% CI 1.04, 2.34, p = 0.030). CHWs provided the highest levels of appropriate treatment compared to other providers. However, improvements in appropriate treatment were observed at health facilities and pharmacies, with CHW appropriate treatment the same across the arms. The rate of CHW attrition in both intervention arms was less than half that of the control arm; adjusted risk difference mHealth arm -4.42% (95% CI -8.54, -0.29, p = 0.037) and VHC arm -4.75% (95% CI -8.74, -0.76, p = 0.021). Appropriate treatment by CHWs was encouragingly high across arms. The inSCALE mHealth and VHC interventions have the potential to reduce CHW attrition and improve the care quality for sick children, but not through improved CHW management as we had hypothesised.

Trial Registration: ClinicalTrials.gov (NCT01972321).

## Author summary

There is global consensus that trained and equipped community health workers (CHWs) are a cornerstone of primary health care and pandemic preparedness. Yet, efforts to improve their motivation and quality of care delivered, in particularly to children with potentially life-threatening conditions like malaria, pneumonia and diarrhoea, are still largely undocumented. This is the first study that has implemented mHealth and community engagement interventions at scale to support community health workers and appropriate treatment of sick children. Our results suggest that interventions that promote learning and increase CHW's confidence and connectedness to the wider health system and community, be it by using mHealth or community engagement, can increase appropriate treatment for malaria, diarrhoea, and pneumonia. However, sub-analysis showed between arm effects were at the level of the health facility and pharmacy with CHWs offering elevated levels of appropriate treatment across arms; possibly due to increased awareness of appropriate treatment among both caregivers and health providers. While the body of evidence on mHealth is increasing, few studies have to date documented the effect on health outcomes. Our robust evaluation, using a cluster randomised controlled trial, is therefore one of the very few that have shown a statistically significant impact of an mHealth intervention on appropriate treatment of sick children. The second intervention, village health clubs (VHCs) also increased appropriate treatment of sick children, in particular for diarrhoea.

## Introduction

Every year 1.8 million children under the age of 5 years die from malaria, diarrhoea, and pneumonia (MDP), accounting for 30% of all-cause mortality in this age group. In 2015 it was projected that a further 21 million children will die from these three treatable conditions by 2030 [1]; a number likely to be further augmented by the COVID-19 pandemic [2]. The ability of many low-income countries to meet the Sustainable Development Goal (SDG) of reducing under 5 mortality is frustrated by sparse numbers of health workers to provide essential services. Integrated Community Case Management (iCCM), endorsed by WHO and UNICEF, utilises community health workers (CHWs) to diagnose and treat MDP at a community level [3]. Evaluations across four African countries, the Democratic Republic of Congo, Malawi, Niger, and

Nigeria, estimated that 6200 children were saved from 2013–16 due to the introduction of iCCM [4]. However, large scale robust evaluations have shown no effect on child mortality, often attributed to suboptimal programme implementation including lack of systems for ensuring the availability of medicines, CHW performance and community engagement [5]. If iCCM, and other CHW programmes are to reach their full potential, there is an urgent need for strategies and innovations that improve performance, supervision, and motivation of CHWs.

Behavioural theories have demonstrated how the processes that determine an individual's behaviour, such as performance, are dependent upon interpersonal relationships and group memberships and their perceived value and significance to the individual [6]. As per this so called "social identity approach" CHWs are more likely to be motivated to perform their duties if their aspirations are perceived to be achievable through the fulfilment of their duties. Thus, interventions that seek to align collective worker goals with the aims of programmes have promise [6]. However, there is limited evidence for how to deliver such interventions at scale.

There is modest, but growing evidence that other CHW focussed interventions, such as mHealth, can improve CHW performance by increasing adherence to treatment recommendations [7,8]. Studies have also shown that mHealth tools for CHWs, such as smartphones programmed with training materials and decision support algorithms, can modestly improve quality of care through better assessment of the sick children and better referral decisions [9] and increase performance feedback and counselling of CHWs [10]. However, robust evaluations of direct impact of mHealth on health and economic outcomes are lacking, as is the evidence on the effectiveness of mHealth delivered at scale [11]. Another potential means of improving CHW motivation and performance is through community engagement. Low utilisation of CHWs by community members, resulting from a lack of buy-in, trust and alignment between CHWs and their communities, has previously been shown to undermine the potential impact of iCCM [5,12]. Therefore, community empowerment approaches, where communities actively participate in activities and decisions that affect their own health through community groups including CHWs have been recommended to improve health seeking behaviours [13].

The Innovations at Scale for Community Access and Lasting Effects (inSCALE) project set out to develop and evaluate two interventions based on the social identity approach (described above) through extensive formative research in Uganda; a mHealth intervention and a participatory community engagement approach delivered through Village Health Clubs (VHCs) [6]. Both were designed to strengthen community health services by enhancing the supervision, motivation, performance, and retention of CHWs to increase the appropriate treatment of childhood MDP. This paper presents the results from a cluster randomised controlled trial (cRCT) in Uganda, which assessed the impact of the mHealth and VHCs interventions compared to control.

## Methods

### Study design and participants

The inSCALE Uganda evaluation was a three-arm cRCT where the unit of randomisation was the sub-county (cluster). The study was conducted across 39 predominantly rural sub-counties in eight districts in the Midwestern region of Uganda. We excluded sub-counties which contained less than 10 villages or where there were no iCCM trained CHWs. In total, 13 sub-counties were randomly assigned to the mHealth intervention, 13 to the Village Health clubs and 13 to usual care (control). All CHWs residing in the study sub-countries were automatically included in the intervention. The study area spanned 24,450 km$^2$ with a population of 1.8 million residing in around 4,000 villages. A detailed study protocol has been published previously [14]. iCCM training had been provided to all participating CHW's by the Non-Governmental

Organisation (NGO) Malaria Consortium and national iCCM trainers from July 2010-June 2011.

## Data collection procedures

Baseline and follow up data were collected using two cross sectional household and CHW surveys across the trial sub-counties. The baseline survey was carried out in June-August 2011. The interventions were implemented from November 2012 to February 2014. In February and March 2013, a mixed methods process evaluation was conducted to understand experiences with intervention uptake and perceived impact, the mechanisms behind CHW motivation and identification with the CHW collective, and to learn about how and why the interventions worked or did not work. The follow-up household and CHW surveys were administered in April-June 2014, 17 months after intervention rollout. Both data collection rounds took place during the end of the rainy season/early dry season. Five villages were randomly selected per sub-county for the household surveys. A list of all households in each community was supplied by the local parish council and verified by the field supervisor. From this list households were selected and surveyed at random, in accordance with the sample size requirements for the evaluation [14].

Household questionnaires, based on standardised questions from the Ugandan Demographic and Health Survey (DHS) and the Multiple Indicator Cluster Survey (MICS) instruments, were administered by trained field staff to the primary caretaker of children under 5 [15,16]. They included socio-economic and demographic characteristics of households, symptoms of illness for children under 5 years in the preceding two and four weeks (at baseline and endline, respectively), care seeking behaviour, treatments received and costs of care seeking. Pictures of locally available drugs for common childhood illnesses were used to increase the accuracy of recall of treatments received for sick children. The questionnaires were extensively field tested and delivered in four local languages. To ensure consistency, 10% of household interviews were repeated by field supervisors within a week of the original and discrepancies resolved. Economic data was collected from a range of sources on the costs of implementing the interventions to estimate the cost effectiveness.

CHW motivation and social identification measurement scales were adapted for use with CHWs based on formative research and theory from the literature [6,17]. Likert scale items relating to motivation were developed and social identification items were adapted for the Ugandan CHW setting [18,19]. The validation process for both scales is published [6] and briefly described in the 'statistical analysis' section below.

## Randomisation and masking

Restricted randomisation was conducted by the study statistician (SS) and occurred after baseline data collection using a programme written in Stata 13 (StataCorp, Texas USA) [20]. The restricted randomisation ensured that between arms at baseline, there was i) no more than a 5 percentage point difference in appropriate treatment for MDP (separately), ii) no more than a 0.5 unit difference in mean log cost of care seeking for a sick child and iii) no more than a 0.5 unit difference in average CHW motivation score [14].

## Interventions

All CHWs across interventions and control sites had previously been trained in iCCM and were equipped with commodities by the Ministry of Health with support from Malaria Consortium; training and commodity provision was identical across arms. iCCM training enabled CHWs in all sites to diagnose, treat and refer sick children presenting with symptoms of MDP

Table 1. Intervention package description.

|  | Control arm | mHealth arm | VHC arm |
|---|---|---|---|
| **Training** | • Basic training in health promotion and health education<br>• Training in iCCM | • Basic training in health promotion and health education<br>• Training in iCCM<br>• VHTs and supervisors trained in 'inSCALE Mobile VHT system' to submit weekly patient data and receive motivational feedback messages. | • Basic training in health promotion and health education<br>• Training in iCCM<br>• VHTs trained as club facilitators to encourage club members to plan and carry out the club's activities using an action and planning cycle. |
| **Accessories and materials** | • Supply of commodities | • Supply of commodities<br>• Nokia C2-00 (Java enabled dual sim card feature phone)<br>• Solar lamp (Sun King Pro) with multiple phone charging pins<br>• Job aids | • Supply of commodities<br>• Picture cards for ranking common child health problems Instructional VHC flip books<br>• T-shirts<br>• Membership cards<br>• Stamps and other stationary to help setting up and running the clubs in the communities |
| **Supervision and support** | • Monthly supervision | • Monthly supervision<br>• Automated SMS sent to supervisors flagging problems and strengths identified in the data submitted, and alerting supervisors about VHTs requiring targeted supervision.<br>• VHT supervisors trained in effective supervision skills using paper based core competency assessment tools and as trainers of VHTs in the mHealth intervention.<br>• Closed user groups for free calls between VHTs and with their supervisors | Monthly supervision<br>• Village leaders appointed as patrons and sensitised to support the mobilisation of the communities to join the clubs.<br>• VHT supervisors, health assistants and sub-county development officers trained in effective supervision skills using a core competency assessment tool and as trainers of VHTs in the VHC intervention. |
| **Reporting tools** | • Paper based VHT register<br>• Monthly paper-based aggregated reports | • Paper based VHT register<br>• Monthly paper-based aggregated reports<br>• 'inSCALE Mobile VHT system' to send aggregated weekly reports on patients seen (sex, mRDT results, symptoms and classification of signs, treatment given and outcome of treatment) and current drug stock levels. | • Paper based VHT register<br>• Monthly paper-based aggregated reports |
| **VHT motivation** | • Financial reimbursement for transport costs for meetings/supervision | • Financial reimbursement for transport costs for meetings/supervision<br>• Relevant and personalised feedback messages based on submitted data sent by SMS instantly after reports are received. | • Financial reimbursement for transport costs for meetings/supervision |
| **Numbers of users** | • 1,012 VHTs in 13 sub-counties in eight districts in the Mid-western region. | • 1,275 VHTs in 13 sub-counties in eight districts in the Mid-western region. | • 880 VHTs across the eight districts, to facilitate the set-up of 440 VHCs. |

using artemisinin combination therapy, oral rehydration salts (ORS) plus zinc or amoxycillin. The interventions were designed based on formative research and social identity theory and have been previously described in detail and are outlined in Table 1 [6,14,19].

In brief, the mHealth intervention encompassed the provision of inSCALE branded phones and solar chargers to CHWs in the mHealth clusters. There were three intervention components: 1) Closed user groups allowing free mobile phone communication among CHWs, and between CHWs and their supervisors; 2) weekly CHW data submission using phones with instant automated performance related feedback, messages to supervisors flagging problems for targeted supervision, and summary statistics made accessible online to district statisticians; and 3) monthly motivational SMS to CHWs with performance improvement tips on good sick child management, and to CHW supervisors on good supervision practices. The overall aim was to promote quality care and to increase CHWs confidence and connectedness to the wider health system using real-time data.

Sixteen Master Trainers trained 49 'trainers-of-trainers' to deliver over 50 three-day courses to 1,277 CHWs and their 37 health facility supervisors on the mHealth intervention. This

cascade model used existing health system training structures and personnel. CHW supervisors were trained on targeted supportive supervision techniques and on how to solve technical problems with hardware and software. CHWs and their supervisors were provided with troubleshooting guides. MOH (Ministry of Health) district biostatisticians were trained to download, review, and respond to data submissions from the CHW phones to detect drug stockouts or unusual data trends.

The community engagement intervention took the form of village health clubs (VHCs) as a platform for participatory and locally owned identification of health problems and solutions, with the CHW as its focal point [6]. The VHCs utilised a four-stage participatory learning and action cycle, where the CHW facilitating the clubs engaged community members to meet monthly to: 1) identify, discuss, and rank local child health challenges, 2) discuss and plan solutions, 3) take planned actions to meet identified challenges, and 4) monitor, report and communicate on their progress. Guiding principles of VHC implementation were that they be open to all in the community, be fun and focused on community health improvement through utilisation of local assets with a particular focus on the CHWs.

In total, eight master trainers trained 39 trainers to deliver 45 CHW training sessions with 884 CHWs (two per village). Training involved adult learning approaches and outlined the VHC approach and materials. Each CHW was equipped with a flipbook and a set of child illness picture cards to facilitate question and answer sessions; a facilitator's starter kit of membership cards, stationery, certificates, and t-shirts; and evaluation forms and attendance registers. Peer to peer learning meetings with other CHWs took place at five and nine months to encourage early adoption sites to share experiences with late adopters on best practices in forming and running the VHCs.

CHWs providing iCCM based care in the control area were provided with regular refresher trainings according to the standard national CHW programme curriculum, including quarterly meetings in health facilities.

## Outcomes

The primary outcome was the percentage of illness episodes (MDP) treated with the appropriate medicine according to WHO informed national guidelines in children aged 2 months to 5 years. Full definitions of suspected MDP and their treatments are provided in S2 File. Secondary outcomes were: i) the percentage of sick children taken to a CHW (first port of call, or at any point for the most recent illness); ii) the percentage of children referred to the next level of care due to stock-outs of essential items at the CHW; iii) mean CHW motivation and social identification scores; and iv) mean CHW clinical knowledge score. Additional trial outcomes were v) the percentage of children with fever who received a rapid diagnostic test for malaria; vi) the percentage of MDP episodes treated both appropriately and within a day of illness onset; vii) mean sub-domain scores for CHW motivation and clinical knowledge; and viii) appropriate treatment of a sick child or "whole child analysis", where each child was categorised as appropriately treated if they received the correct treatment for all reported episodes of MDP occurring the last 4 weeks and not appropriately treated if they were not correctly treated for one or more episode occurring in that period.

## Sample size estimation

Sample sizes were based on the primary outcome, namely appropriate treatment of cases with suspected malaria, diarrhoea, and pneumonia. Procedures for calculating sample size are described in full in the study protocol paper [14]. In brief, assuming a control group percentage of appropriate treatment of 47% episodes of MDP and a coefficient of variation of 0.17

(values estimated from baseline data) gave us 90% power to detect a 15% change in appropriate treatment as a result of either intervention, sampling 155 children in each of 39 clusters. This sample size would also enable us to detect changes of 15%-25% in appropriate treatment of reported malaria, diarrhoea, and pneumonia individually with power of at least 80% [14].

## Statistical analysis

All analyses were intention-to-treat and accounted for the cluster-randomised design. This was achieved by inclusion of the variables on which the randomisation was restricted at baseline as fixed effects (appropriate treatment for MDP, health worker motivation and household log cost of care seeking), and specifying random-intercepts for sub-county cluster units for either binary or continuous outcomes.

For binary outcomes odds ratios were converted to relative risks (RR) using the marginal standardisation technique, and the 95% CIs of the RRs estimated with the delta method [21]. Sensitivity analyses of appropriate treatment (primary outcome) were conducted to account for alternative treatment definitions and statistical models (difference-in-differences and adjustment for baseline variables used in restricted randomisation). Analysis of the trial was conducted initially on a version of the results dataset which masked sub-county allocation; once analysis methods were agreed by the study team, a final unmasked impact analysis was performed.

To assess the impact of the interventions on CHW motivation and social identification, scores were generated from the validated CHW motivation and social identification measurement scales. To validate the scales, qualitative, psychometric factor analysis, and item-reduction methods were applied. These methods have been published elsewhere [6]. This analysis resulted in CHW work motivation scores estimated based on a three-factor scale (general motivation, retention in role and reward for effort) and social identification scores estimated based on a single factor, four item scale that explained the majority of variation in social identification between CHWs. The items in the final CHW work motivation ($\alpha$ = 0.67) and social identification ($\alpha$ = 0.69) tools showed good internal consistency.

The standardised CHW clinical knowledge score was created as the weighted sum of each correct procedure mentioned by the CHW in response to a scenario. Specific questions and individual weights are provided in the S1 File.

## Process evaluation

In February and March 2013, a mixed methods process evaluation with 1874 self-administered quantitative CHW checklists and 64 qualitative key informant interviews with CHWs, their supervisors and district leaders were conducted. For the mHealth arm, questions included the number of CHWs who could make calls in the closed user groups, submit weekly data, received monthly motivational SMS, feedback messages after data submission, or phone calls from their supervisor. For the VHCs arm, questions included number of villages that had formed, and were still running a VHC, number and frequency of VHC meetings held and VHC attendance, and reasons for low or slow adoption of VHCs. Open ended questions were asked on attitudes about and perceptions of the interventions.

The aim was to understand intervention uptake and perceived impact, the mechanisms behind CHW motivation and social identification with the CHW collective, and how and why the interventions worked or did not work. Quantitative data were entered in Excel and analysed descriptively using frequencies. Qualitative interviews were digitally audio-recorded, transcribed verbatim into English, and manually coded and analysed using a thematic content analysis approach. Intervention fidelity is presented below; full details of the qualitative component of the process evaluation have been published elsewhere [6].

## Ethics

The inSCALE trial, registered as NCT01972321, was approved by Makerere University Institutional Review Board, the Uganda National Council of Science and Technology (ref. HS 958), and London School of Hygiene & Tropical Medicine Ethics Committee in the UK (ref. 5762). Oral consent for the random allocation of districts to intervention or control groups was obtained from the sub-county leadership. Individual written informed consent was obtained from the caregiver for baseline and follow up data collection after explaining the purpose of the interview. Participants were free to decline the interview at any time.

## Results

Fig 1 shows the trial profile. Baseline data showed that there was balance between arms for key demographic characteristics, including primary caretaker type, age of respondent, education level completed, marital status, job type, and religion, with some differences found in ethnicity and mother tongue spoken, of which neither were seen to be related to the outcome (Table 2). Baseline appropriate treatment rates of diarrhoea, pneumonia and malaria were not significantly different in the three arms. Roughly 47% of children with any episode of MDP received appropriate treatment (from any provider) during their illness.

At follow-up, 7679 households were surveyed, and 3413 households were found to have at least one child under 5 years. From these households, caregivers were asked about illness episodes for a total of 5573 children aged 2 months to 5 years; 1858 in the control arm, 1858 in the community engagement arm and 1856 in the mHealth arm. Out of these children, 946 (control arm), 979 (VHC arm) and 881 (mHealth arm) had had MDP symptoms in the last one month, representing 51%, 53% and 48% respectively (Table 3). Background characteristics of the children with MDP at follow-up is outlined in Table 4.

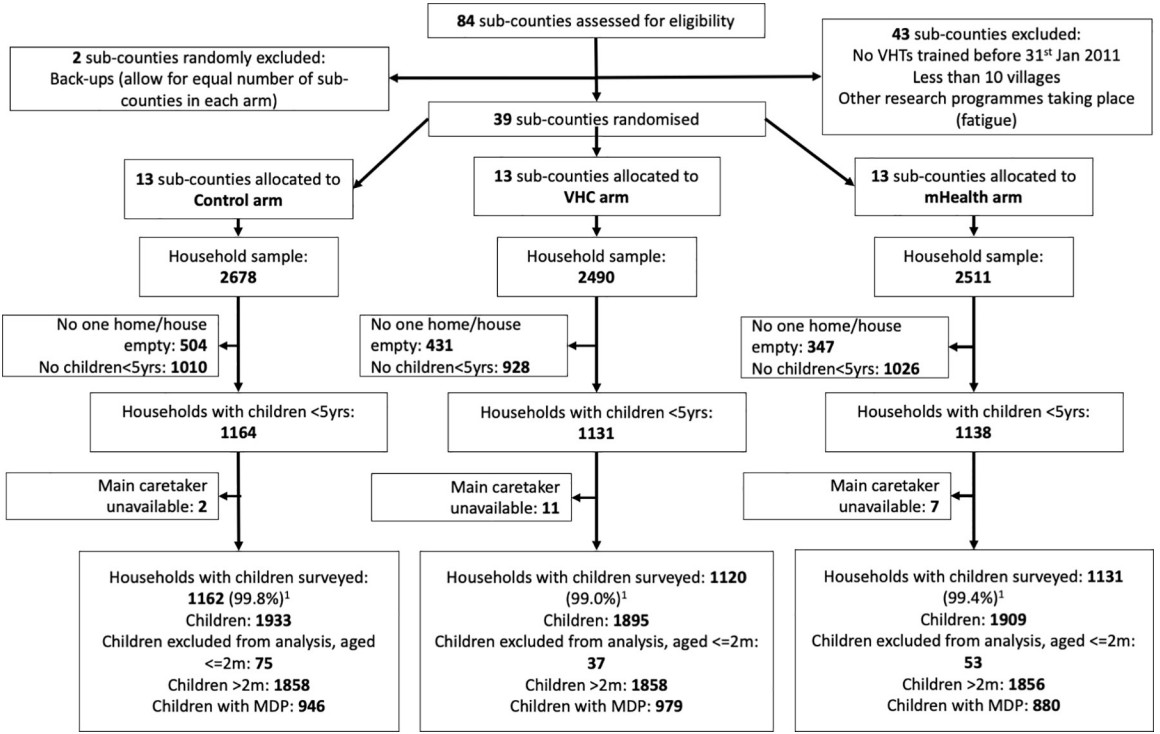

**Fig 1. Flow diagram of participant progression through trial phases.**

**Table 2.  Baseline comparability of inSCALE intervention arms after restricted randomisation.**

| Outcome | Control (n = 13 clusters) | VHC (n = 13 clusters) | | mHealth (n = 13 clusters) | |
|---|---|---|---|---|---|
| | Cluster mean (sd) | Cluster mean (sd) | Community/Control ratio | Cluster mean (sd) | mHealth/ Control ratio |
| % appropriate treatment for MDP | 48% (11.4) | 47% (9.1) | 0.97 | 49% (6.5) | 1.03 |
| % appropriate treatment for fever | 48% (13.1) | 46% (12.6) | 0.96 | 48% (7.8) | 0.99 |
| % appropriate treatment for diarrhoea | 31% (16.9) | 30% (15.1) | 0.96 | 30% (13.2) | 0.97 |
| % appropriate treatment for suspected pneumonia | 54% (13.0) | 53% (12.3) | 0.99 | 58% (8.2) | 1.08 |
| CHW motivation | 2.3 (0.4) | 2.3 (0.4) | 1.03 | 2.5 (0.5) | 1.08 |
| Household cost ($Log_{10}$) of treatment for children with MDP | 3.1 (0.2) | 3.2 (0.3) | 1.01 | 3.1 (0.2) | 1.00 |

**Table 3.  Prevalence of suspected malaria, diarrhoea, or pneumonia (MDP) at endline.**

| Characteristic | Control (n = 1858) | VHC (n = 1858) | mHealth (n = 1856) |
|---|---|---|---|
| Prevalence of MDP | 50.9% (946) | 52.7% (979) | 47.5% (881) |
| Suspected malaria | 46.8% (842) | 48.4% (868) | 43.9% (787) |
| Diarrhoea | 10.6% (197) | 13.1% (244) | 8.5% (158) |
| Suspected pneumonia | 14.1% (262) | 15.3% (284) | 12.5% (231) |

**Table 4.  Characteristics of inSCALE participants (children with MDP) at follow-up by intervention arm.**

| Characteristic | Control (n = 946) | VHC (n = 979) | mHealth (n = 880) |
|---|---|---|---|
| % respondents are mother of sick child (n) | 70% (659) | 59% (581) | 62% (548) |
| % boys (n) | 52% (492) | 53% (522) | 52% (461) |
| Age of child (months) mean (sd) | 29.4 (15.7) | 28.4 (15.3) | 28.2 (15.7) |
| Age of respondent (years) mean (sd)[1] | 33.8 (12.4) | 34.3 (12.5) | 35.0 (13.3) |
| Highest education level completed % (n) | | | |
| None | 23% (217) | 24% (233) | 19% (171) |
| Incomplete primary | 46% (436) | 49% (484) | 49% (427) |
| Primary or higher | 31% (293) | 27% (262) | 32% (282) |
| Ethnicity % (n) | | | |
| Luo | 22% (208) | 44% (430) | 16% (137) |
| Runyoro-rutoro/Nkore-kiga | 50% (477) | 43% (425) | 52% (460) |
| Luganda | 9% (81) | 1% (11) | 25% (223) |
| Other | 19% (180) | 12% (113) | 25% (223) |
| Marital status % (n) | | | |
| Married | 60% (569) | 70% (681) | 62% (542) |
| Living together | 27% (255) | 19% (183) | 21% (180) |
| Single/divorced/widowed | 12.8% (121) | 11.8% (115) | 18.0% (158) |
| % farmer/fisher(wo)man | 80% (760) | 83% (810) | 81% (711) |

[1]Age unknown in 3% (25) of control, 3% (30) community and 2% (17) technology arm respondent

**Table 5. Treatment and care seeking outcomes by intervention arm.**

| Outcome | Control arm | VHC arm | | | mHealth arm | | |
|---|---|---|---|---|---|---|---|
| | % (n/N) | % (n/N) | RR vs control (95% CI) | p | % (n/N) | RR vs control (95% CI) | p |
| Primary outcome | | | | | | | |
| % episodes MDP appropriately treated | 62% (804/1301) | 68% (944/1396) | 1.08 (1.00–1.18) | 0.056 | 69% (808/1176) | 1.10 (1.01–1.20) | 0.035 |
| Secondary outcomes | | | | | | | |
| % children with MDP first seeking care to CHW | 38% (356/942) | 27% (267/976) | 0.79 (0.56–1.11) | 0.167 | 39% (338/878) | 1.15 (0.83–1.59) | 0.409 |
| % children with MDP seeking care to CHW at any point | 38% (360/942) | 28% (269/976) | 0.74 (0525–1.04) | 0.084 | 39% (342/878) | 1.06 (0.77–1.47) | 0.715 |
| % children with MDP referred by CHW due to drug stock outs[1] | 11% (36/318) | 15% (39/258) | 1.41 (0.76–2.60) | 0.274 | 10% (33/317) | 0.95 (0.48–1.90) | 0.882 |

*The logistic regression model included a random effect to model variation between clusters, and fixed effects for baseline rates of appropriate treatment for fever, diarrhoea and pneumonia, baseline CHW motivation score, baseline rate of CHW care-seeking and baseline log mean cost of household care-seeking.

Table 5 summarises the analyses of the primary outcomes for the two interventions. Sick children in the mHealth arm were significantly more likely to receive appropriate treatment for MDP compared with those in the control arm (RR 1.10; 95% CI 1.01, 1.20; p = 0.035). Though not statistically significant, the largest effect size was seen on appropriate treatment for diarrhoea, with a risk ratio of 1.39 for providing ORS (95% CI 0.90–2.15; p = 0.085) and 1.28 for providing ORS plus zinc (95% CI 0.70, 2.36; p = 0.424). There was an 8% increase in appropriate treatment in the VHC intervention arm for appropriate treatment of MDP in children, though again this did not reach statistical significance at the 5% level (RR 1.08; 95% CI 1.00, 1.18; p = 0.056). As with the mHealth arm, the largest effect was seen in appropriate treatment for diarrhoea, with a significant risk ratio of 1.54 (95% CI 1.12, 2.12; p = 0.007). These findings are consistent with the results of a "whole child" concurrent symptoms analysis (S3 File) where children were categorised as appropriately treated only if all their symptoms were treated correctly (i.e., if they had one or more of MDP concurrently).

The inSCALE interventions did not increase community utilisation of the CHW as a care provider for MDP, or care seeking in general (Table 5). However, care-seeking from CHWs increased over time across all three arms; compared to baseline the CHW care seeking in the control arm increased by nearly 15% points at endline. The increase in CHW care seeking was less in the VHC arm than in the mHealth and control arms (S4 File).

No differences were noted in work motivation or knowledge of CHW following the introduction of inSCALE interventions, and CHW work motivation and social identification was notably high across all arms (Table 6). However, the rate of CHW attrition in both

**Table 6. CHW retention, motivation, social identification, and performance by intervention arm.**

| Outcome | Control arm | VHC arm | | | mHealth arm | | |
|---|---|---|---|---|---|---|---|
| | Mean (sd) | Mean (sd) | Y-X difference (95% CI) | p | Mean (sd) | Y-X difference (95% CI) | p |
| Mean CHW motivation score (min = 25; max = 77) | 67.2 (5.28) | 68.35 (5.40) | 1.22 (-0.98, 3.42) | 0.277 | 68.01 (5.34) | 0.59 (-1.33, 3.25) | 0.410 |
| Mean CHW social identification score (min = 4; max = 20) | 17.95 (1.97) | 17.82 (1.60) | -0 .08 (-0.71, 0.54) | 0.794 | 18.42 (1.66) | 0.61 (-0.04, 1.26) | 0.066 |
| Mean CHW knowledge score (min = 0; max = 56) | 50.0 (12.43) | 48.3 (8.74) | -1.17 (-6.92, 4.59) | 0.691 | 47.9 (11.1) | -2.02 (-7.98, 3.93) | 0.506 |
| Mean CHW attrition (%) (min = 0; max = 100) | 7.8 (7.3) | 3.2 (3.9) | -4.75 (-8.74, -0.76) | 0.021 | 4.6 (3.7) | -4.42 (-8.54, -0.29) | 0.037 |

**Table 7. Proportion of episodes of suspected malaria, diarrhoea or suspected pneumonia that were appropriately treated in each intervention arm; by first care seeking location and symptom group.**

| Outcome | Control arm | VHC arm | | | mHealth arm | | |
|---|---|---|---|---|---|---|---|
| | Freq (%) | Freq (%) | Risk ratio versus control (95% CI) | p | Freq (%) | Risk ratio versus control (95% CI) | p |
| First care seeking location | | | | | | | |
| CHW | 74% (352/478) | 76% (285/374) | 1.03 (0.95–1.12) | 0.444 | 79% (357/455) | 1.03 (0.95–1.12) | 0.450 |
| Public Facility | 66% (187/284) | 72% (317/440) | 1.06 (0.95–1.20) | 0.285 | 78% (194/248) | 1.13 (0.99–1.28) | 0.064 |
| Private facility | 55% (191/345) | 64% (256/400) | 1.13 (0.98–1.29) | 0.092 | 56% (170/305) | 1.01 (0.83–1.14) | 0.742 |
| Pharmacy | 50% (44/88) | 61% (55/90) | 1.27 (0.95–1.70) | 0.107 | 66% (63/95) | 1.44 (1.08–1.90) | 0.012 |
| General shop /herbalist/other | 60% (9/15) | 44% (10/23) | 0.91 (0.39–2.13) | 0.823 | 42% (5/12) | 0.69 (0.28–1.71) | 0.418 |
| Did not seek care outside the home | 23% (21/91) | 30% (21/69) | 1.35 (0.72–2.53) | 0.355 | 31% (19/61) | 1.34 (0.70–2.59) | 0.375 |
| Symptom group | | | | | | | |
| Suspected malaria | 74% (624/842) | 78% (679/868) | 1.05 (0.99–1.12) | 0.089 | 80% (630/787) | 1.07 (1.01–1.14) | 0.024 |
| Confirmed malaria | 88% (401/458) | 88% (437/499) | 1.00 (0.95–1.05) | 0.952 | 91% (426/470) | 1.02 (0.97–1.07) | 0.504 |
| Diarrhoea (ORS) | 29% (58/197) | 47% (114/244) | 1.54 (1.12–2.12) | 0.007 | 40% (63/158) | 1.37 (0.96–1.955) | 0.085 |
| Diarrhoea (ORS + zinc) | 13% (26/197) | 17% (41/244) | 1.29 (0.77–2.14) | 0.334 | 17% (26/158) | 1.32 (0.75–2.34) | 0.334 |
| Suspected pneumonia | 47% (122/262) | 53% (151/284) | 1.15 (0.89–1.49) | 0.298 | 50% (115/231) | 1.05 (0.78–1.40) | 0.758 |

intervention arms was less than half that of the control arm over the 18 months period from initial CHW hiring and training (December 2012) to the end of follow-up (April-June 2014) after adjusting for baseline differences: 7.8% attrition rate in the control arm compared to 3.2% in the VHC arm (adjusted risk difference: -4.75%, 95% CI -8.74, -0.76, p = 0.021) and to 4.6% in the mHealth arm (adjusted risk difference: -4.42%, 95% CI -8.54, -0.29, p = 0.037).

CHWs provided the highest level of appropriate treatment across care providers in all arms of the study (Table 6) e.g., CHW'S provided on average 18% higher rates of appropriate treatment compared to private providers across arms. However, though unexpected, it is important to note that, the overall proportion of children with MDP appropriately treated by a CHW in the mHealth arm (RR 1.05; 95% CI 0.96, 1.16; p = 0.286) and VHC arm (RR 1.04; 95% CI 0.94, 1.14; p = 0.481) did not differ significantly with children treated by CHWs in the control arm. In contrast, there was a trend towards more children receiving appropriate treatment in the intervention arms for most providers apart from general shops or herbalists; though this only reached significance for children taken first to pharmacies in the mHealth arm (Table 7). Children in the VHC arm treated in private facilities were more likely to receive appropriate treatment than in private facilities in the control arm.

Based on CHW responses, intervention fidelity was moderate, with the majority of CHWs in the mHealth arm (83%) still had a functioning phone at endline survey, but only 38% had submitted reports in the last month. In the VHC arm, 94% of villages had held at least one VHC meeting and 81% of villages having organised more than three meetings in the past year (Table 8). Monitoring and process evaluation data showed that several software and hardware problems occurred in the mHealth arm during the implementation, limiting the reporting frequency. However, following system rectification half-way through the implementation 60% of CHWs were able to submit their reports on a weekly basis. In addition, most of the other

**Table 8. Intervention fidelity in the inSCALE arms.**

| Fidelity indicator—VHC arm | Outcome |
| --- | --- |
| % villages held at least one VHC meeting between April 2013 –April 2014 (n/N) | 94% (59/63) |
| % CHWs organised 3+ VHC meetings between April 2013 –April 2014[T] (n/N) | 81% (47/58) |
| % organised 6+ VHC meetings between April 2013 –April 2014 [T] (n/N) | 54% (31/58) |
| Mean number of VHC meetings organised (incl no meetings) (sd) | 7 (6.02) |
| Mean number of participants at most recent meeting (sd) | 34.2 (28.3) |
| Intervention fidelity–Technology arm | |
| % CHWs had a working inSCALE mobile phone at endline survey (n/N) | 83% (85/102) |
| Mean number of reports submitted with phone in last 4 weeks | 38% (39/102) |

[T]Information on number of VHCs not available for 4 villages

intervention components, such as the closed user groups and monthly SMS feedback messages, were functional for most users throughout the implementation period. While it was anticipated that CHWs reporting on drug stock levels in the technology arms may encourage re-distribution between CHWs to reduce stock-outs, national CHWs policies prevented health facilities to act on to drug stock information.

## Discussion

This is the first study that has implemented and evaluated the impact of mHealth and community engagement village health clubs (VHCs) to support delivery of iCCM. Our results suggest that the mHealth intervention increased appropriate treatment for malaria, diarrhoea, and pneumonia (MDP) compared with children in the control arm. However, contrary to our hypothesis, the mechanism of this improvement was not directly through the CHWs; rather the improvement seen in the mHealth intervention was due to increased levels of appropriate treatment at public health facilities and pharmacies. In a recent review of the use of digital health for strengthening health systems, Long et al (2018) [11] showed that solutions for health worker training, provider-to-provider communication and professional networking, and supervision of and performance feedback to health workers may lead to better health workforce development. While the body of evidence on mHealth is increasing, leading to WHO publishing its first global recommendations on digital interventions for health system strengthening [8], few studies have to date been able to document an effect on health outcomes [9,22]. Our robust evaluation, using a randomised controlled trial, is therefore one of the very few that have shown a statistically significant impact of an mHealth intervention on appropriate treatment of sick children.

The VHCs also increased appropriate treatment of MDP in children by 8%, though the improvement did not reach statistical significance at the 5% level. However, sub-group analysis showed that children in the VHC arm were more often treated appropriately for diarrhoea with ORS than those in the control group, and the effect was large (56%) and statistically significant. The only other study published on the effect of village health clubs on child health outcomes showed no effect on diarrhoea in children in Rwanda [23]. In our study, there was minor difference in diarrhoea prevalence between arms but there was a shift in providers sought when children were ill, with more children in the VHC arm being taken to public lower-level facilities compared to in the other arms.

While we cannot conclude that the two inSCALE interventions improved use of CHWs by community members, nor appropriate treatment provided by CHWs, care seeking from CHWs did increase over time across all three arms. This effect was less prominent in the VHC

arm, as more children in this arm resorted to public health facilities. It is difficult to explain what caused the documented positive effects on appropriate treatment of sick children, as neither of the CHW focused interventions seem to have increased knowledge, motivation or appropriate treatment provided by CHWs directly. Challenges related to the iCCM programme (e.g. irregular drug supply, low CHW literacy and visual impairment) as well as to the specific interventions (e.g. lack of network connectivity, software and hardware maintenance, and partnerships with software partners in the mHealth arm, and lack of interest from the community, inadequate stationery supplies, and insufficient support from local leaders in the VHC arm) likely limited the impact of the interventions [24,25]. However, it is possible that these types of health systems strengthening interventions, delivered through cascade training using facility based health workers, sensitisation and support to health facility supervisors and other district stakeholders, have led to general system improvements; the exact nature and mechanism of this improvement needs further exploration in different health contexts and implementation studies. From the caregiver interviews during the process evaluation, caregivers in the VHC arm frequently mentioned that the VHC sessions have taught them that pneumonia and diarrhoea are potentially severe conditions that need urgent care, which could explain why more caregivers resort to health facilities in this arm.

In 2016 WHO released the recommendation on implementation of community mobilization through facilitated participatory learning and action (PLA) cycles with women's groups to improve maternal and newborn health, particularly in rural settings with low access to health services [26]. In line with findings from our study, the report state that the pathways of influence on outcomes are difficult to assess, but that that women meeting to identify their needs and seek solutions plays a vital role; other activities related to the solutions identified at the meetings may also play a role. Three years later, in 2019, WHO released a guideline on recommendations on digital interventions for health system strengthening [8]. Three of the nine evidenced based digital interventions recommended were partially embedded in our mHealth arm, namely 'notification of stocks and commodities' using mobile devices to communicate using e.g. text messaging (SMS) and data dashboards to manage and report on supply levels, 'digital provision of training and educational content to health workers' using channels such as SMS text messaging, and 'provider-to-provider telemedicine' where less skilled health workers are linked with more specialist ones for a variety of reasons, including to get assistance with diagnoses and to conduct case-management consultations. As also concluded in our study, the guideline points out that researchers should be realistic about the extent to which digital health interventions can impact on distal health outcomes, which are often affected by a variety of factors beyond the interaction with the digital intervention, including access to medicines, their cost, family support, and biomedical factors. It is therefore suggested that health system focused digital interventions, with large complexity in the number of components, behaviours targeted, and organizational levels involved, may make designs such as randomized controlled trials for evaluating the effectiveness of these interventions difficult to apply, and other designs may therefore need to be considered [8].

CHW work motivation and social identification, which had not previously been measured in low-income settings, were high across and within intervention arms. High reported motivation is consistent with recent results in Uganda and other comparable settings [27], though may have been subject to social desirability bias despite attempts during scale development to avoid this. Elsewhere it has been suggested that CHW motivation should only be measured when adequate working conditions, such as, appropriate supervision and supply of drugs have been provided [28]. While these were controlled for in the design of the inSCALE trial, in practice they were not always in place. Potentially the interventions led to a CHW perception that adequate working conditions had been provided in intervention areas which contributed to

the lower attrition rates. High reported CHW social identification, indicating shared goals and commitment to the collective, suggest that further investigation of the performance priorities of an apparently motivated CHW workforce utilising this theoretical lens has explanatory promise.

Our study limitations included the delivery of the interventions in only one area of Uganda limiting study generalisability across contexts. While contamination between study arms is possible, it was likely minimal, given that community members tend to use the CHW who they have elected for their community. In addition, the implementation, though working with the Ministry of Health, was heavily supported by Malaria Consortium, an experienced and highly motivated NGO. Evaluation of similar interventions delivered solely by government staff would be useful to consider what it would take to sustain the interventions at scale. Appropriate treatment was assessed by parental recall, as previously done in national and international surveys [15,16], rather than objective notes audit. This method was the only feasible data collection method due to the location and scale of the project. The delay in publication of these findings was due to organisational dispersal of the original study team. However, the size and robust nature of the study makes it relevant especially with the need for mobilisation and strengthening of the health workforce in Sub-Saharan Africa during the current COVID-19 pandemic.

The inSCALE interventions increased appropriate treatment of children with MDP in Uganda and reduced CHW attrition rate by half. CHW delivered iCCM resulted in encouragingly elevated levels of appropriate treatment of childhood MDP across intervention and control arms. The improvements observed in the intervention arms occurred despite programmatic challenges such as drug stockouts and irregular supervision; barriers which these interventions could not overcome. Alternative pathways, such as care seeking from public and private health facilities, was the main contribution to the improvement. We recommend that future studies aiming to evaluate drivers of CHW performance and motivation further try to unpack the mechanisms of action, as well as the care seeking pathways among caregivers who did and did not seek care from a CHW, to understand their motivations. We also see a lot of value in evaluating a combination of aspects from both the VHC and mHealth intervention, as these approaches might have complementary benefits. A combined VHC-mHealth intervention would also more closely reflect the reality of today's CHW programmes, which increasingly adopt digital health tools for CHW task management, data reporting and eLearning, in combination with community mobilisation activities to optimise CHW management.

## Supporting information

**S1 File. Scoring instrument for inSCALE CHW Performance.**
(DOCX)

**S2 File. Illness and appropriate treatment definitions for fever, diarrhoea and pneumonia.**
(DOCX)

**S3 File. Appropriate treatment results based on whole child analysis.**
(DOCX)

**S4 File. Baseline-endline change in appropriate treatment and CHW care seeking across inSCALE arms (difference-in-difference analysis).**
(DOCX)

## Acknowledgments

We want to thank the research assistants who collected the data, the inSCALE team Uganda and the participants of the study for their time and insight.

## Author Contributions

**Conceptualization:** Karin Källander, Seyi Soremekun, Daniel Ll Strachan, Zelee Hill, Sylvia Meek, James Tibenderana, Betty Kirkwood.

**Data curation:** Seyi Soremekun, Guus Ten Asbroek, Patrick Lumumba.

**Formal analysis:** Seyi Soremekun, Frida Kasteng, Guus Ten Asbroek, Anna Vassall, Raghu Lingam, Betty Kirkwood.

**Funding acquisition:** Karin Källander, Sylvia Meek, James Tibenderana.

**Investigation:** Karin Källander, Seyi Soremekun, Daniel Ll Strachan, Zelee Hill, Frida Kasteng, Edmound Kertho, Agnes Nanyonjo, Guus Ten Asbroek, Maureen Nakirunda, Patrick Lumumba, Godfrey Ayebale, Benson Bagorogoza, Anna Vassall, James Tibenderana, Raghu Lingam, Betty Kirkwood.

**Methodology:** Karin Källander, Seyi Soremekun, Daniel Ll Strachan, Zelee Hill, Frida Kasteng, Guus Ten Asbroek, Anna Vassall, Sylvia Meek, Raghu Lingam, Betty Kirkwood.

**Project administration:** Karin Källander, Edmound Kertho, Maureen Nakirunda, Patrick Lumumba, Godfrey Ayebale, Sylvia Meek, James Tibenderana.

**Resources:** Sylvia Meek.

**Supervision:** Karin Källander, Zelee Hill, Edmound Kertho, Godfrey Ayebale, Anna Vassall, Sylvia Meek, James Tibenderana, Raghu Lingam, Betty Kirkwood.

**Validation:** Zelee Hill, Agnes Nanyonjo, Anna Vassall, Raghu Lingam, Betty Kirkwood.

**Writing – original draft:** Karin Källander, Seyi Soremekun, Raghu Lingam.

**Writing – review & editing:** Karin Källander, Seyi Soremekun, Daniel Ll Strachan, Zelee Hill, Frida Kasteng, Edmound Kertho, Agnes Nanyonjo, Guus Ten Asbroek, Maureen Nakirunda, Patrick Lumumba, Godfrey Ayebale, Benson Bagorogoza, Anna Vassall, James Tibenderana, Raghu Lingam, Betty Kirkwood.

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
