## [Decision Letter · Decision Letter 0]

30 Aug 2022

PDIG-D-21-00112

Improving community health worker treatment for malaria, diarrhoea and pneumonia in Uganda through inSCALE community and mHealth innovations: A cluster randomised controlled trial

PLOS Digital Health

Dear Dr. Källander,

Thank you for submitting your manuscript to PLOS Digital Health. After careful consideration, we feel that it has merit but does not fully meet PLOS Digital Health's publication criteria as it currently stands. Therefore, we invite you to submit a revised version of the manuscript that addresses the points raised during the review process.

The reviewers specifically highlighted the need for providing more detailed descriptions of the intervention and the statistical analyses. In addition, the discussion of the process evaluation should be expanded. The detailed comments are attached.

Please submit your revised manuscript within 60 days Oct 29 2022 11:59PM. If you will need more time than this to complete your revisions, please reply to this message or contact the journal office at digitalhealth@plos.org. Please include the following items when submitting your revised manuscript:

We look forward to receiving your revised manuscript.

Kind regards,

Laura M. König

Academic Editor

PLOS Digital Health

Journal Requirements:

1. Please update your online Competing Interests statement. If you have no competing interests to declare, please state: “The authors have declared that no competing interests exist.”

2. Please amend your online detailed Financial Disclosure statement. This is published with the article. It must therefore be completed in full sentences and contain the exact wording you wish to be published.

Please state the initials, alongside each funding source, of each author to receive each grant.

3. Please provide separate figure files in .tif or .eps format and remove any figures embedded in your manuscript file. Please also ensure that all files are under our size limit of 10MB.

For more information about how to convert your figure files please see our guidelines: https://journals.plos.org/digitalhealth/s/figures

4. Please add a full list of legends for all your Supporting Information files after the references list.

5. Thank you for submitting your data to a repository. We have been unable to access the data using the link/accession number you provided. Please contact the repository to ensure that the links and accession numbers are valid. If a link or accession number needs updating, please provide a revised Data Availability Statement including the information.

Additional Editor Comments (if provided):

Reviewers' comments:

Reviewer's Responses to Questions

**Comments to the Author**

1. Does this manuscript meet PLOS Digital Health’s publication criteria? Is the manuscript technically sound, and do the data support the conclusions? The manuscript must describe methodologically and ethically rigorous research with conclusions that are appropriately drawn based on the data presented.

Reviewer #1: Yes

Reviewer #2: Yes

2. Has the statistical analysis been performed appropriately and rigorously?

Reviewer #1: No

Reviewer #2: I don't know

3. Have the authors made all data underlying the findings in their manuscript fully available (please refer to the Data Availability Statement at the start of the manuscript PDF file)?

Reviewer #1: Yes

Reviewer #2: Yes

4. Is the manuscript presented in an intelligible fashion and written in standard English?

Reviewer #1: Yes

Reviewer #2: Yes

5. Review Comments to the Author

Reviewer #1: This is cluster randomized trial conducted in Uganda where the authors reported that The inSCALE mHealth and VHC interventions have the potential to reduce CHW attrition and improve the care quality for sick children. However, there are some caveats that need to be addressed.

1. The sample size was known only after the study was done. Can the authors provide the sample size calculation with different ingredients used for its estimate? 

2. In the methods section

a. There was a suggestion of using only random effect model (Statistical analysis, line 280) for the multivariate model. However, in table 5, both random and fixed effect were used. 

b. Also, can the authors provide rational of using intention to treat analysis (line 277, statistical analysis) and not per protocol or both?

3. In the limitation section (page 28), one will wonder if contamination cases between study arms were not found.

4. On page 28, last two sentences, it is mentioned that the improvements observed in the intervention arms occurred despite problems with drug stocks and other programmatic challenges. Can the author specify the type of programmatic challenges?

5. Also in the conclusion section, one would expect to see the study perspectives or future. What can the authors suggest from this study in that regard? 

6. Line 267, there was an typos error.

Reviewer #2: This manuscript presented the results of a large cluster randomized trial. Importantly, process evaluation results are presented alongside the main trial findings. While this increases the manuscripts complexity, this combination helps the reader to more fully understand the trial results in context. My primary comment lies in the description of the interventions and trial arms. Other comments that strengthen results reporting are also listed below.

Abstract – the number of groups is not quite clear in the abstract. Says two interventions, but later on there are three arms (mHealth; VHC or usual care (control) arms)? It is understood once the reader reads the paper, but I think it should be more clear up front.

Mechanism of improved CHW management presented in Interpretation is not well referenced in the Results section

Introduction

Paragraph starting on line 127 describes the two approaches to increasing CHW engagement and effectiveness and they are presented as individual approaches. However, it is unclear to this reader why these two approaches are not complementary or could not be combined into a package approach. Meaning, what was the rationale for testing these approaches separately rather than together? Furthermore, Line 146 seems to indicate they have the same potential effects.

Line 143 The Innovations at Scale for Community Access and Lasting Effects (inSCALE) project set out to develop and evaluate two interventions based on the social identity 

Please clarify the phrase “…based on the social identity…”

Methods

Table 1. Unclear what is meant by the phrase “and patrons” in the phrase “Supervisor support and patrons”. 

Some overlap, such as supervisor support among the entries. Suggest review of each row and merging them (not separating by mHealth solutions, village health clubs, CHW program support)

Page 17, line 291: This is not clear – is this referring to the scales introduced in line 195? Why were these methods applied to the scales used? 

Results

Fig 1, please add abbreviations under table. Suggest match Arm names the names in the manuscript for consistency

Page 22 – consider adding “whole child” concept to Methods section

Provide scale ranges for all variables as footnotes (e.g., Table 6) 

Discussion 

I would appreciate a larger discussion of process evaluation that you would consider adding to future work to elucidate some of the unknowns behind the mechanism of action. For example, interviews with caregivers who did and did not seek care to understand their motivations

6. PLOS authors have the option to publish the peer review history of their article (what does this mean?). If published, this will include your full peer review and any attached files.

**Do you want your identity to be public for this peer review?** For information about this choice, including consent withdrawal, please see our Privacy Policy.

Reviewer #1: No

Reviewer #2: No

---

## [Decision Letter · Decision Letter 1]

13 Dec 2022

PDIG-D-21-00112R1

Improving community health worker treatment for malaria, diarrhoea and pneumonia in Uganda through inSCALE community and mHealth innovations: A cluster randomised controlled trial

PLOS Digital Health

Dear Dr. Källander,

Thank you for submitting your manuscript to PLOS Digital Health. Before we can accept the manuscript for publication, please address the remaining comment by Reviewer 2.

Please submit your revised manuscript within 30 days Feb 11 2023 11:59PM. If you will need more time than this to complete your revisions, please reply to this message or contact the journal office at digitalhealth@plos.org. Please include the following items when submitting your revised manuscript:

We look forward to receiving your revised manuscript.

Kind regards,

Laura M. König

Academic Editor

PLOS Digital Health

Journal Requirements:

2. Please send a completed 'Competing Interests' statement, including any COIs declared by your co-authors. If you have no competing interests to declare, please state "The authors have declared that no competing interests exist". Otherwise please declare all competing interests beginning with the statement "I have read the journal's policy and the authors of this manuscript have the following competing interests:"

3. Please amend your detailed Financial Disclosure statement. This is published with the article. It must therefore be completed in full sentences and contain the exact wording you wish to be published.

4. We have noticed that you have uploaded Supporting Information files, but you have not included a full list of legends. Please add a full list of legends for your Supporting Information files after the references list.

Additional Editor Comments (if provided):

Reviewers' comments:

Reviewer's Responses to Questions

**Comments to the Author**

1. If the authors have adequately addressed your comments raised in a previous round of review and you feel that this manuscript is now acceptable for publication, you may indicate that here to bypass the “Comments to the Author” section, enter your conflict of interest statement in the “Confidential to Editor” section, and submit your "Accept" recommendation.

Reviewer #1: All comments have been addressed

Reviewer #2: All comments have been addressed

2. Does this manuscript meet PLOS Digital Health’s publication criteria? Is the manuscript technically sound, and do the data support the conclusions? The manuscript must describe methodologically and ethically rigorous research with conclusions that are appropriately drawn based on the data presented.

Reviewer #1: Yes

Reviewer #2: Yes

3. Has the statistical analysis been performed appropriately and rigorously?

Reviewer #1: Yes

Reviewer #2: Yes

4. Have the authors made all data underlying the findings in their manuscript fully available (please refer to the Data Availability Statement at the start of the manuscript PDF file)?

Reviewer #1: Yes

Reviewer #2: Yes

5. Is the manuscript presented in an intelligible fashion and written in standard English?

Reviewer #1: Yes

Reviewer #2: Yes

6. Review Comments to the Author

Reviewer #1: (No Response)

Reviewer #2: The authors have done a sufficient job responding to both sets of reviewers' comments. However, there is one in which I still have a comment:

Provide scale ranges for all variables as footnotes (e.g., Table 6)

Authors’ response: We have now provided the scale ranges for fixed-score-based outcomes in Table 6 (Page 23 Results)

The request was to provide possible scale ranges for all variables as footnotes, e.g., motivation score 1 (low) to 10 (high). I believe the authors have added the maximum observed value in the table 5 itself.

7. PLOS authors have the option to publish the peer review history of their article (what does this mean?). If published, this will include your full peer review and any attached files.

**Do you want your identity to be public for this peer review?** For information about this choice, including consent withdrawal, please see our Privacy Policy. 

Reviewer #1: Yes: 

Reviewer #2: No

---

## [Decision Letter · Decision Letter 2]

17 Feb 2023

Improving community health worker treatment for malaria, diarrhoea and pneumonia in Uganda through inSCALE community and mHealth innovations: A cluster randomised controlled trial

PDIG-D-21-00112R2

Dear Dr Källander,

We are pleased to inform you that your manuscript 'Improving community health worker treatment for malaria, diarrhoea and pneumonia in Uganda through inSCALE community and mHealth innovations: A cluster randomised controlled trial' has been provisionally accepted for publication in PLOS Digital Health.

Best regards,

Laura M. König

Academic Editor

PLOS Digital Health

Reviewer Comments (if any, and for reference):

Reviewer's Responses to Questions

**Comments to the Author**

1. If the authors have adequately addressed your comments raised in a previous round of review and you feel that this manuscript is now acceptable for publication, you may indicate that here to bypass the “Comments to the Author” section, enter your conflict of interest statement in the “Confidential to Editor” section, and submit your "Accept" recommendation.

Reviewer #2: (No Response)

2. Does this manuscript meet PLOS Digital Health’s publication criteria? Is the manuscript technically sound, and do the data support the conclusions? The manuscript must describe methodologically and ethically rigorous research with conclusions that are appropriately drawn based on the data presented.

Reviewer #2: (No Response)

3. Has the statistical analysis been performed appropriately and rigorously?

Reviewer #2: (No Response)

4. Have the authors made all data underlying the findings in their manuscript fully available (please refer to the Data Availability Statement at the start of the manuscript PDF file)?

Reviewer #2: (No Response)

5. Is the manuscript presented in an intelligible fashion and written in standard English?

Reviewer #2: (No Response)

6. Review Comments to the Author

Reviewer #2: (No Response)

7. PLOS authors have the option to publish the peer review history of their article (what does this mean?). If published, this will include your full peer review and any attached files.

**Do you want your identity to be public for this peer review?** For information about this choice, including consent withdrawal, please see our Privacy Policy.

Reviewer #2: None
